# Menu Engineering and Dietary Behavior Impact on Young Adults’ Kilocalorie Choice

**DOI:** 10.3390/nu13072329

**Published:** 2021-07-07

**Authors:** Christine Bergman, Yuan Tian, Andrew Moreo, Carola Raab

**Affiliations:** 1Food & Beverage and Event Management Department, University of Nevada Las Vegas, Las Vegas, NV 89119, USA; cherrytian1128@163.com (Y.T.); Carola.Raab@UNLV.EDU (C.R.); 2Chaplin School of Hospitality Management, Florida International University, Miami, FL 33181, USA; AMoreo@FIU.EDU

**Keywords:** nutrition labeling, menu, menu engineering, young adults, calories, obesity, stages of change

## Abstract

The obesity pandemic is associated with increased consumption of restaurant food. Labeling of menus is an intervention used to provide consumers with kilocalorie (calorie) information in hopes of them making healthier food choices. This study evaluated the relationship between young adults’ calorie choices on restaurant menus and menu design, dietary behaviors, and demographic characteristics. A 3 (fast-casual restaurants) × 4 (menu-designs based on menu engineering theories) between-subjects (n = 480, 18–24-year olds) experimental design was used. The relationship between the participants’ calorie choices (high versus low) and menu design, stage of change, gender, race, educational level and weight status was evaluated using logistic regression. All independent variables had at least one category that had greater odds (CI 95% ± 5%) of subjects choosing a lower calorie entree, except education level and race/ethnic group. Normal weight and overweight subjects had greater odds of choosing lower calorie entrees than those that were obese. In addition, subjects that had started to control their calorie intake for less than six months or had sustained this change for at least six months, had greater odds of choosing lower calorie entrees compared to others. Including a green symbol and calories on fast casual restaurant menus may influence some young adults to choose lower calorie entrees.

## 1. Introduction

The obesity pandemic has a well-documented effect on chronic disease risk and, as a result, costs the worlds’ economies an immense amount of money [1,2,3,4]. Eating food from restaurants reportedly increases the risk of becoming overweight and obese, the evidence is strongest for fast-food-types of restaurants. Compared to dining at home, the foods sold at restaurants typically have larger portion sizes, are lower in fiber, and higher in kilocalories (calories), saturated fat, cholesterol, and sodium [5,6,7]. For example, the average adult in the U.S. consumes an additional 143 calories per day and gains two pounds each year by consuming food from restaurants once a week [8,9].

In 2007, money spent on food purchased away-from-home (i.e., all restaurant-types) was, for the first time in the U.S., more than that spent on food at home [10]. By 2018, purchases on food away-from-home increased to 54% of total food expenditures. Similar increases in spending on food-away-from-home have occurred globally, primarily in urban areas [11]. For example, in 2011, urban households in the U.S. spent 28% more on food away-from-home than rural households [12]. The section of the world’s human population living in urban regions is predicted to grow from 55% in 2018 to 60% in 2030 [13]. Therefore, the percentage of food consumed by humans that is prepared at restaurants, will likely continue to increase globally [14].

A report by the McKinsey Global Institute [2] suggested several interventions that should be considered for inclusion in an integrated societal response to the obesity epidemic. One such intervention was to provide consumers with food labeling that would allow them to understand the nutrient profile of their food and/or its calorie content. The U.S., Canada, Australia, Saudi Arabia, Ireland, Taiwan, South Korea, and the United Arab Emirates require most restaurants to provide customers with nutritional information on menus [15]. The most common requirement is that calories be provided on menus, menu boards, or drive through signs.

There have been several reviews and a meta-analysis that examined the effects of including nutrition information on adult food choices from menus [16,17,18,19,20,21]. The general consensus from these reviews is that calorie labels on menus have the desired effect on menu choices in some contexts (i.e., fewer calories chosen), but no trends on effectiveness have been established in terms of subject demographics, types of restaurant, or menu design. In terms of menu labeling formats, study subjects noticed and understood color codes (e.g., red traffic light = high, amber traffic light = medium, and green traffic light = low calorie) and physical energy equivalents compared to calories posted on menus [18]. The most recent meta-analysis studied the effect of calorie labeling on menus in laboratory and in away-from-home settings [21]. A significant effect was found in the former studies, but not in the latter ones. However, the authors of that study moderated the significance of their conclusions because few studies were included in their analyses.

Seaberg [22] first introduced the concept of menu design psychology, and various menu design proposals, also known as menu engineering. Since their introduction, they have been examined and, in certain instances, found to offer restaurateurs methods to promote specific menu items [22,23]. A study by Robertson and Lunn [24] used menu engineering to evaluate hypotheses of how menu design may influence the calorie content of consumers’ menu choices. These authors reported that placing price and calorie information next to menu items was associated with consumers ordering foods that had 93 fewer calories (11%) in comparison to a control group. The effect was largest when the calorie content was in the same font and placed on menus to the right of the price.

Some common characteristics of previous studies on menu labeling are that data were not collected on subjects’ lifestyle and health behaviors [25]. As stated by Sinclair et al. [26], “menu labeling research is in its infancy.” It has yet to include designs that test the health literacy of menu labeling interventions among diverse population subgroups. Therefore, the purpose of this study was to evaluate the relationship between young adults’ calorie choices on fast-casual restaurant menus and various factors, including menu designs, personal dietary behaviors, body mass index, degree of hunger, and demographic characteristics. The study hypothesis is as follows: young adults will be positively influenced by several menu engineering formats to choose lower calorie entrees from a simulated fast-casual restaurant menu. In addition, several demographic and physical characteristics of young adults will also be associated with them choosing lower calorie entree choices.

## 2. Materials and Methods

The research methods for this study included a survey and an experiment. U.S. adults who were 18 to 24 years old (*n* = 490) were the subjects. A sample size of 500 reportedly provides an acceptable level of statistical power when evaluating data using the primary analytical method used for this study, specifically logistic regression [27]. In addition, the smallest sample size for an acceptable level of statistical power using logistic regression was calculated post hoc and found to be N = 10k/*p* = 10*17/36.3% ≈ 468, in this case p is the proportion of the respondents that chose lower-than-600 calorie items, and k is the independent variables [27]. Thus, the power for this study was acceptable.

The participants were contacted using an online research panel from a pre-arranged pool of respondents who had agreed to be contacted by the Qualtrics company. The subjects were all in the U.S., had Internet access, reported to have eaten in a fast-casual restaurant at least once in the previous six months.

For validity and reliability purposes, between subject design was used so that participants were not in the experimental group and the control group. This study was approved by the UNLV Office of Research Integrity Human Subjects (protocol #724286-2).

The experiment utilized a 3 (restaurant type) × 4 (menu design) between-subjects design, which resulted in the creation of 12 menus (Appendix A). Subjects were parsed according to their preference for a restaurant type, in order to limit the data variance in restaurant preference, increase the study’s internal validity, and produce a better estimate of the treatment effects.

The survey consisted of two subsections: (1) the menu selection section and (2) multiple-choice questions section. Participants were asked to choose their preferred restaurant from the following fast-casual choices: “Urban Mexican”, “Asian Fusion”, or “California Café”. One of the menus with a different design was randomly picked by the software and presented to the participant. They were then asked to choose one item from the “lunch special entrée” section. A selection from the beverage section was allowed but not included in the analysis.

All the calorie information, menu items and descriptions were obtained from three fast-casual restaurant chains, considered the models for this simulation study. The “Asian Fusion” entrees and calorie contents (i.e., calories ranged from 365 to 1060) were based on a menu from Pei Wei Asian Diner; the “California Café” entrees (i.e., calories ranged from 510 to 1140) were based on the menu from Panera Bread, and the “Urban Mexican” entrees (i.e., calories ranged from 310 to 1190) were based on a menu from Baja Fresh Mexican Grill. These fast-casual restaurants are required by Title 21 Code of Federal Regulations 101.11(c) to list the calorie content of foods on menus [28]. The calorie values were determined using nutrient databases, nutritional software, laboratory analyses, cookbooks, and other reasonable means. The exact method of nutrient value determination does not need to be provided to customers.

Three theories of menu design psychology were used to develop the menu designs, specifically, the serial position effect, the gaze motion theory, and salience builder effect [29,30,31]. In brief these theories are as follows: place highly-profitable menu items in the top-right corner of the menu, the place some diners reportedly first look; people more accurately recall the first and last items of a list compared to other items; and distractions from default preferences such as icons next to words in a list can catch people’s attention, respectively.

Each entree item on the menus were labeled with the exact calorie content from the model restaurant to the right of its name. The menus each had 8 entrees with higher-than-600 calories and 4 entrees with lower-than-600 calories. The calorie cut off of 600 was used as it is approximately the amount 1/3 (i.e., 1 meal out of 3) of the 2000 calories used to calculate the % daily values required on U.S. food labels [28]. To allow for personal preferences in addition to restaurant-type, all menus provided three pork/ham dishes, three vegetarian dishes, three chicken/turkey dishes, and three beef/steak dishes; each category of dishes included 1 lower-than-600 calorie dish and 2 higher-than-600 calorie dishes.

Standardized 1-page-layout design was applied to all the 12 menus, in terms of calorie information format, menu color, font size and color, orientation, menu size, price of lunch specials and beverages, and price format. The menu designs used in the study had no similarity to the format of the menus from the reference restaurant, mentioned above. Identical prices in the entree sections were applied in all menus in order to control the bias in the results that price might produce. In addition, all menus contained the following statement at the bottom: “A 2000 Calorie daily diet is used as the basis for general nutrition advice, however, individual needs may vary” [28].

The 3 menu treatments included “first and last” menus designed to have the 4 lower-than-600 calorie items located at the top and bottom of the two menu columns; the “light and fresh” menus were designed with a greenlight symbol next to lower-than-600 calorie items; and the “sweet spot” menus were designed to have 4 lower-than-600 calorie items at the upper right of the 2 column menus. Control menus looked just like the treatment menus including placement of calories next to each entree name and the entree items were listed randomly throughout the menu.

The transtheoretical model (TTM) and its stages of change developed by Prochaska, Redding, and Evers [32] were used to identify participant’s stage related to any of their efforts to change their weight. Specifically, participants were asked to answer multiple-choice questions about their recent dietary behavior and any behavior change, which were based on the staging instrument developed by Curry, Kristal, and Bowen [33]. Based on the answers to these questions, subjects were put into one of the following categories: precontemplation, contemplation, preparation, action, and maintenance (Table 1). Past studies found a significant association between diet self-ratings and independent measures of fat intake (*p* < 0.001), thus suggesting sufficient validity for its use [34]. The present study adapted the instrument by exchanging the word fat with calories.

Each participant was also asked questions about their sex, education, race/ethnicity, height/weight and current degree of hunger. The education categories used were as follows: high school or equivalent, some college, vocational training/technical, associate’s degree, bachelor’s degree, master’s degree or higher. The race/ethnicity groups offered to the subjects as choices were Asian/Pacific Islander, African American (or Black), Hispanic (or Latino), multiracial, Native American or White. “Body Mass Index (BMI) was calculated using each person’s height and weight. The formula is BMI = kg/m^2^ where kg is a person’s weight in kilograms and m^2^ is their height in meters squared.” [1]. The following are categories nominalized from BMI values: underweight if the BMI is below 18.5, normal (i.e., healthy weight) if 18.5 to 24.9, overweight if the BMI is between 25 and 29.9, and obese if the BMI is equal to or higher than 30. Subjects were presented with the statement, I am hungry right now, and asked to rate their hunger using a Likert scale with the following choices: strongly disagree, disagree, undecided, agree, and strongly agree.

Before starting the study, the instrument was examined by four professionals, well versed in survey research and experimental design: two were familiar with the study’s menu engineering topic and two were not. The study was explained and they were asked to evaluate the instrument for question clarity and to determine if they measure what they were designed to measure. Changes were made to the survey based on the reviewers’ comments to establish face validity. A group of 28 university undergraduate students (ages between 18 and 24 years old) that responded to an email request to help with a research project also examined the survey. They were asked to take the survey and make comments on the clarity and comprehensibility of its directions and questions. Changes were made to the survey based on the comments from the students prior to the launch of the survey. In addition to performing these validation activities, an internal consistency check was performed by including 2 questions in different places in the survey that were related to participants’ education level, but were formatted and worded differently. Subjects that did not answer these 2 questions equivalently were dropped from the dataset while it was cleaned by 2 authors at different times.

The binary dependent variable in this study was subjects’ choice of an entree that was < or ≥600 calories. The independent variables for this study included: restaurant type, menu design, BMI category, stage of change, gender, race, and education. Restaurant-type included those described above as “Asian Fusion”, “California Café”, and “Urban Mexican”. Menu-design included those described above as the control (no menu engineering treatment), first and last, light and fresh, and sweet spot. The BMI categories of underweight, healthy weight, overweight, and obese were used as described above. Stage of change data categories used were pre-contemplation, contemplation, preparation, action, or maintenance, as described above [33]. Asian and Pacific Islander, African American or Black, Hispanic or Latino, Multiracial, Native American, and White were the race/ethnic groups used. Education was categorized into 7 groups: no high school, high school or equivalent, some college credit or no advanced degree, vocational or training/technical school, associate degree, bachelor’s degree, and master’s degree.

International Business Machines (IBM, Armonk, NY, USA) SPSS Statistics for Windows, Version 24.0. was used to perform descriptive statistics of the subjects and cross tabulations between the independent and dependent variables. In addition, binary logistic regression was used to model the probability of subjects choosing the outcome variable which was an entree with < or ≥600 calories [35].

Before conducting the logistic regression, collinearity diagnostics were run in order to detect any redundant variables. Specifically, variance inflation factor (VIF) and tolerance values were run using all of the independent variables, the results were all < 2 and <0.1, respectively, indicating collinearity was not a concern [36].

A purposeful model building technique was used. Specifically, univariable analysis was performed first to examine the association between the independent variables and the outcome variable [37]. Independent variables with a *p* value of <0.25 were then used for multivariable logistic regression analysis. The final step was to evaluate the model goodness of fit using the Hosmer–Lemeshow statistic.

The simple contrast function was used to compare each type of the menu-type independent variable (i.e., First and Last, Light and Fresh, and Sweet Spot) to the control menu (i.e., reference). Sex (female, male, or other) was also evaluated using the simple contrast. As nominal variables, the deviation contrast was applied to race/ethnic group as independent variables in the logistic regression model in order to compare participants in different race/ethnic groups to the overall mean value. The difference (i.e., reverse Helmert) contrast was used to compare each level of the independent variables, BMI, stages of change, education and degree of hunger to the previous category in the model (except the first one).

Only 6 people out of 23 that received the “first and last” menu from the California Café restaurant chose the lower-than-600 calorie items. Of those participants that selected the Asian Fusion restaurant, only 10 out of 40 people that received the control menu and 10 out of 37 people that received the “light and fresh” menus chose the lower calorie entree. Consequently, according to Peduzzi et al. [27] the regression coefficients could be biased in both positive and negative directions with low cell numbers (≤10). Therefore, the logistic regression analyses did not include restaurant type within the models to ensure each cell count was more than 10, as recommended for logistic regression analysis [27].

## 3. Results

### 3.1. Demographics

Data collection resulted in 480 complete responses collected from young adults aged 18 to 24 years old. However, 9 subjects’ responses were deleted from the dataset due to evidence of that their answers were entered randomly, that is their answers did not pass the internal consistency check, resulting in an *n* = 471 (Table 2).

The subject group was compromised of more males (57.7%) than females (41.2%) (Table 2). The top four groups of the subjects in terms of their race/ethnicity were Caucasian (66.9%), Asian (13%), Hispanic or Latino (7.4%) or African American or black (7.2%). More of the subjects had a bachelor’s degree (34.2%) than any of the other levels or types of education. More than 50% of the subjects were in the normal BMI category and approximately one quarter were categorized as obese.

On the contrary, the average education level among subjects was greater than the national level (Table 2) [38]. Almost half of the participants have an associate degree, bachelor’s degree and master’s degree whereas about 80% of the U.S. young adult population have a high school degree and/or are currently attending colleges.

### 3.2. Restaurant-Type

After picking their restaurant choice at the onset of the survey, the subjects were randomly presented with 1 of 4 menus under the chosen restaurant type, thus ensuring all menus were presented to approximately the same number of participants (Table 3). “Urban Mexican” was the most popular restaurant type among the participants (*n* = 45.2%). The “California Café” was the least popular restaurant type (*n* = 22.3%), while the “Asian Fusion” restaurant was selected by 32.5% of the participants.

### 3.3. Menu Design and Calorie Choice

As shown in Table 4, fewer subjects chose menu items with <600 calories (*n* = 36.3%) compared to those that picked an item with >600 calories (*n* = 63.7%). The simple contrast function was applied in a logistic regression model in order to compare each menu design to the control menu. In comparison to the control menu, the “light and fresh” menu (i.e., green symbol placed next to the lower-than-600 calorie items) resulted in a significant increase in the portion of subjects who chose entrees lower in calories (B = +0.552, Sig. = 0.04), however, none of the other menu designs had a significant effect on the choice of menu items.

### 3.4. Stages of Change and Calorie Choice

The transtheoretical model was applied to categorize people into five stages of behavioral change according to their dietary behavior and behavior change, and operationalized to evaluate the correlation between personal dietary behavior and young adults’ calorie choices on restaurant menus. The distribution of participants in the five stages of change were as follows: 48% in pre-contemplation, 7.1% in contemplation, 18% in preparation, 18% in action, and 8.9% in maintenance (Table 4). Most participants were in the pre-contemplation stage, that is, not currently limiting their daily calorie consumption and were not planning to change their dietary behavior in the near future.

There were higher percentages of participants who choose the lower-than-600 calorie items in the 4 stages other than the pre-contemplation stage as compared to the pre-contemplation stage (Table 4). This, indicates that subjects who were more likely to switch to a lower calorie lifestyle and more likely to maintain healthy dietary behaviors, were also more inclined to choose lower calorie items on restaurant menus. The highest percentage of people who chose <600 calorie items were in the action stage, and the second highest in the maintenance stage, indicating that participants who either started to control their calorie amount in their diet over the past six months or less or have sustained this change for at least six months and intend to continue, provided the strongest reactions to the calorie information on the menu.

Logistic regression (95% confidence interval) with the difference contrast function found a significant effect of participants’ dietary behavior change and their calorie choices on restaurant menus (Sig. = 0.05). In addition, in comparison to participants in the preparation stage (stage 3), there was a significant increase in the portion of those choosing lower-than-600 calorie entrees among those in the action stage (stage 4) (B = +0.565, Sig. = 0.03). This suggests that people in the action stage who recently changed to a healthier diet over the past six months or less were significantly more sensitive to the calorie information on the restaurant menus.

### 3.5. Demographics and Calorie Choice

Cross-tabulation results between participants’ gender and their calorie choices on restaurant menus indicated that the percentage of females choosing lower-than-600 calorie items was higher than the male subjects choosing lower-than-600 calorie items (45.9% vs. 30.1%) (Table 4). Logistic regression (95% confidence interval) using the simple contrast function found a significant effect of a participants’ sex on their calorie choices from a restaurant menu (Sig. = 0.003). More females choose lower calorie menu items compared to males (B = +.675, Sig. = 0.001).

Cross tabulation between participants’ educational levels and their calorie choices on restaurant menus found the number that picked a < 600 calorie entree with a high school degree or less, some university, vocational/technical training, associate degree, bachelor’s degree, or master’s degree to be as follows: 34.3%, 30.6%, 45.5%, 38.2%, 39.1%, and 52.9%, respectively (Table 4). Using logistic regression (95% confidence interval) with a difference contrast no significant relationships between educational levels in general and low-calorie items food choices were found (Sig. = 0.395).

The results of cross tabulation analysis between the reported race of the participants and those that choose a menu item of <600 calories were as follows: Asian (32.3%), Black or African American (32.4%), Hispanic or Latino (37.1%), Multiracial (47.8%), Native American or American Indian (0%) and White (36.8%) (Table 4). Logistic regression (95% confidence interval) with a deviation contrast showed no significant relationships between race in general and food choices of <600 calorie entrees (Sig. = 0.85), nor was there a specific race that had significant association with lower calorie choices.

Using cross tabulation analysis an inverse relationship between participants’ BMI and their choice of a low-calorie entree was found (Table 4). That is, a greater percentage of underweight people choose a lower calorie menu items compared to those that had a higher BMI.

The difference contrast function was applied to weight status as the independent variables in a logistic regression model so as to compare each weight status to the one that had a lower range in BMI. The subjects in the normal weight status were significantly (Sig. = 0.02) less likely to choose a lower calorie menu item compared to those in the underweight group. The effect was also significant between the overweight group and the normal weight category (Sig. = 0.05). Thus, as participants BMI increased from the first category to another, they were less likely to choose lower-than-600 calorie items (B = −1.079, −0.595, and −0.122, respectively); except those in the overweight and obese groups which had the same likelihood to choose lower-than-600 calorie items.

### 3.6. Degree of Hunger and Calorie Choice

Cross-tabulation results between participants’ self-reported responses to the statement “I am hungry right” were as follows: Strongly Disagree (33.3%), Disagree (34.2%), Neither Disagree nor Agree (36.8%), Agree (38.5%), Strongly Agree (40.3%) (Table 4). Logistic regression analysis revealed that the effect of the subjects’ self-reported degree of hunger on their likelihood to choose lower-than-600 calorie items was not significant (Sig. = 0.91).

Logistic regression analysis was performed using all of the independent variables except for race, education, and response to the statement “I am hungry right now”, as they all had a *p* > 0.25 in the univariable models (Table 5). The final logistic regression model had levels of significance/non-significance for the independent variables that were similar to those found for the models previously run for each individual independent variable. The Hosmer–Lemeshow statistic indicates a poor fit if the significance value is less than 0.05. [37] The model adequately fits the data as the statistic calculated was 1.0.

## 4. Discussion

According to the U.S. Census Bureau, in 2012, those aged from 18–24 were segmented into the top four groups by race/ethnicity as follows: Caucasian (56%), Asian (5%), Hispanic or Latino (19%) or African American or Black (15%). Therefore, the sample used for this study over represents Caucasians and Asians and underrepresents Hispanics (or Latinos) and African Americans (or Blacks) [38]. In addition, the sample in this study was also different in terms of education level compared to the U.S. population of young adults.

Mandatory and voluntary menu labeling policies have been put in place in various countries, to nudge consumers to decrease their calorie intake and to encourage food industry employees to reformulate menu items to contain fewer calories and more targeted nutrients [15]. This study evaluated the effects of menu engineering, stage of change, and BMI of young adults on the odds of them making a low or high calorie menu choice from a simulated fast-casual restaurant. All of the independent variables had at least 1 category that had significantly greater odds of young adults choosing a lower calorie entree.

The “light and fresh” (i.e., green symbol placed next to the lower-than-600 calorie items) menu resulted in a significant increase in the portion of subjects that chose items lower in calories. These results support the findings of previous studies that determined that traffic lights (i.e., red, amber, and green) with or without calorie numeric information on menus result in consumers choosing fewer calories [39,40,41]. In addition to these findings, the data indicates that only a green light symbol is needed in combination with numeric calorie information to encourage some customers to order a restaurant entree that is less than 1/3 of an adult’s daily caloric needs. This symbol may be preferred by restaurant managers so they can avoid including a red symbol on their menus as it could denotate that some menu items should be avoided.

Neither the “First and Last” or “Sweet Spot” menu designs had an effect on subjects’ choice of entree calorie amount in the present study. Lower calorie items placed at the beginning and the end of each menu column (i.e., “First and Last”) according to the rules of recency and primacy should have been remembered to a greater degree than the other menu items. It is possible that this occurred, but this is irrelevant because it did not translate into subjects choosing more lower calorie entree items. The sweet spot employed in this study was the space above the midpoint of the right side of the page. This area of the menu is commonly referred to as the “focal point” of a menu during menu design, although why it is thought to be the focal point has not been established [22]. Yan [40], in a literature review, discussed that there is a lack of empirical evidence supporting the effect of primacy and recency and a sweet spot on restaurant menu item purchase behavior, purchase intention, actual sales, or even attention. This list can now include that there is no support that these psychological methods are able to influence young adults’ odds of choosing a lower calorie entree from a menu.

A study by Larsen et al. [41] evaluating the snack ordering behavior of young adults, found 57% remembered noting calorie information while purchasing a meal or snack at a restaurant in the previous month. Thus, it is possible that the menu engineering method, “Light and Fresh”, that increased the odds of subjects choosing a lower calorie entree could have a greater effect after a consumer sees the green logo repeatedly over time.

The stage of change the subjects were in was associated with greater odds of choosing a lower calorie entree. Compared with people in the preparation stage (stage 3), there was a significant increase in the odds of people choosing the lower-than-600-calorie items compared to people in the action stage (stage 4). This, indicates that subjects in the action stage who just changed to a lower calorie diet over the past six months or less were significantly more sensitive to the calorie information on restaurant menus. Participants not planning to modify their intake of calories (i.e., pre-contemplation) and those only considering doing so it in the future (i.e., contemplation) were not influenced by calorie information on the menus. This is not surprising since these subjects have had no thoughts or practice in considering the calorie content of their meals. On the other hand, the subjects in the maintenance stage (i.e., currently limiting their calories in their diets and have been doing so for at least six months) may consider eating out at a restaurant as a treat and thus do not allow calories to influence their menu item choice.

No other menu labeling studies could be found to compare the results of the association we found between a person’s stage of change and the calorie content of their entree choice. However, a study in fast food restaurants found significant differences in total calories purchased by people who indicated they used the calorie information and those who did not [41]. It is possible that the people that used the calorie information are those, like in the present study, that were in the action stage of change and recently changed to a lower calorie diet making them more likely to notice and act on the calorie information on the menus. The question then becomes if consumers’ in the action stage of change are indeed influenced by calorie menu labeling to make lower calorie choices at restaurants, how can menus be designed to influence the others? Or what educational campaigns might influence others to follow suite in their menu choices?

No previous studies were found that addressed the association between people’s BMI and the odds of choosing menu items labeled as being relatively low in calories. In the present study, those classified as being normal weight or overweight had greater odds of choosing lower calorie menu items than those that were categorized as being obese. This suggests that posting calories on a menu for fast-casual restaurants may assist some customers to make healthier choices, but not those with the highest BMI. However, because all menus contained calories by the entree this cannot be stated with confidence. That is, a menu containing no calorie information was not included in the study since chain restaurant are required to post calorie content on menus by U.S. Federal Law [8]. The relationship between customer BMI and the calorie content of their menu item choice needs further study.

Results from four nationally representative surveys, found that BMI increased sharply as adolescents moved into young adults in the U.S. [42]. Most young adults in the present study were in the pre-contemplation stage, that is, not currently limiting their daily calorie consumption and were not planning to change their dietary behavior in the near future. Similarly, Walsh et al. [43] and Park et al. [44] reported that young adults were primarily in the pre-action stage (i.e., pre-contemplation) for vegetable consumption. Adults are known to eat less than the recommended number of vegetable servings and this is one of the variables associated with their increased risk of weight gain [45]. When questioned, young adults were more concerned about whether food was grown organically or how processed it was compared to how their diet matched up against dietary recommendations taught in schools, such as five-a-day and MyPlate [46]. Thus, young adults need additional public messages and tools to take them from knowing about what a healthy diet is to translating this into supportive behaviors, such as choosing restaurant items with more vegetables and lower calories.

Education level, race/ethnic group and response to the statement “I am hungry right now” of young adults were also examined in the present study to determine the odds of them choosing a low or high calorie menu item from a simulated fast-casual restaurant. Age and race/ethnic group were not associated with greater odds of the subjects choosing a lower calorie menu item. These results agree with the most recent meta-analysis that evaluated the effects of food labeling on consumers nutritional-related food choices [47]. That study also found that sex had no association with food choices in previous menu labeling studies; which is in opposition to the results of the present study. Specifically, women had greater odds of choosing a lower calorie entree compared to men. These results cannot be compared directly since the present study focused on young adults while the meta-study included all adults of all ages. In addition, the meta-analysis included menu labeling studies that occurred in other venues besides restaurants, such as schools and grocery stores. Thus, it is possible that menu labels impact young adults differently than adults in general. However, more targeted studies are needed to assess this.

Subjects in this study were asked about how hungry they were because hunger is known to influence appetite and thus food choice to some degree. The amount of hunger reported did not change the odds of a subjects’ likelihood of choosing a lower calorie entree. These results are similar to that reported by Reale and Flint [48]. The difference is these authors studied obese adults in a laboratory setting, while the present study evaluated young adults in all BMI categories. Degree of hunger was associated with the calorie amount of food items chosen by females with disordered eating, when selecting food items from menus labeled with the number of calories [49]. These conflicting results are not surprising considering that people sometimes eat when sated and at other times refrain from eating even though they are hungry, for various reasons [50]. Thus, degree of hunger is known to influence humans’ food choices, but it is clear that many other factors influence energy intake as well.

A limitation of this study is that the menu selection was conducted in an online environment instead of in a fast-casual restaurant. The use of the online environment can introduce sampling bias in that all members of the target population do not have equal access to this format. The current study did not record when a subject last ate and relied on self-reporting of existing health behaviors and height and weight, which are also limitations. The population in this study and in other studies using online survey platforms were more educated than the U.S. population overall [35]. In addition, the sample for this study was different from the U.S. population in terms of race.

It is important to note that this study focused on the effects of calorie labeling on restaurant menus, and it is nearly impossible to predict and control personal preference for certain dishes, although this effect was limited using a blocked statistical design. The study adopted BMI as one of the predictors of calorie choices on restaurant menus. Body mass index is calculated from a person’s weight which includes both muscle and fat weight, thus some participants may have a high BMI with a low percentage of fat and thus be a healthy weight [51].

Despite the aforementioned study limitations, the findings highlight that a green symbol should be considered by restaurateurs to denotate lower calorie menu items along with the number of calories. This may encourage some young adults to make healthier choices at fast-casual restaurants. In addition, restaurant managers should consider including in their marketing campaigns that they use a unique label to denote lower calorie menu items. This combined with other marketing efforts to promote the restaurant as being health-friendly would fit with the current trend of consumers desiring healthy dining choices.

Future research should continue to evaluate the effects of menu psychology on customers’ healthy food choices from restaurant menus. Although, only one out of the three menu engineering methods that were evaluated in this study showed efficacy, this does not mean that other novel methods should not be created and studied in the future. In addition, larger cohorts need to be studied to allow additional independent variables and interactions to be studied with enough power to detect other statistically significant effects on customers’ calorie choices.

In this study two of the stages of change categories were not associated with reduced calorie choice. It is possible that a menu engineering technique could be created that assists all people in making lower calorie choices from menus. The alternative is that menu engineering techniques tailored for people that are in the other stages of change related to food choices from menus will need to be developed and displayed along with a green light denoting lower calorie items. Matching appropriate menu interventions to an individual’s readiness to change may be able to assist with their progression of stage and thus consume a healthier level of calories.

Perhaps symbols indicating calorie content of menu items may work better in fast-food and fast-casual food operations because customers are often younger and limited in time, thus decisions must be made quickly. On the contrary, full-service restaurants where adults typically have more time to review the menu may only need calories noted in numeric form to assist customers in making healthier choices.

The results of menu labeling studies on the calorie content of subjects’ food choices have been inconsistent, thus the effectiveness of menu labeling continues to be unclear. In part this is likely due to the inconsistent use of variables such as participants’ degree of hunger, nutrition knowledge, and stage of change that have been examined in these studies. An initiative to create and support a task force of scholars working in this field could be brought together to tease apart the findings to-date on human behavior associated with food choices from restaurants. This group could then be tasked with developing a strategic plan for characterizing the interactions among the factors (i.e., demographic, health risk, lifestyle, occupational, behavioral, cultural, economic, and environmental) that may influence the calorie content of peoples’ food choices in restaurants. Strategies to understand the barriers to making lower calorie menu item food choices should also be created.

Government policy should not rely solely on menu labeling to influence all consumers to make healthier food choices in restaurants. Additional research is also needed to determine what other public health initiatives will nudge people to make better food choices, such as healthy eating marketing campaigns and price support of fruit and vegetable production in order to reduce point of sale costs for restaurants and consumers.

## 5. Conclusions

This study evaluated the effect of menu engineering, dietary behavior and various demographics on the calorie content of entrees chosen by young adults in a simulated fast-casual restaurant. Green symbols along with numeric values on menus were associated with young adults that made lower calorie entree choices. Subjects who just changed to a healthier diet over the past six months or less had greater odds of choosing lower calorie food items; the same was true for those with a lower BMI. These results from this simulation study need to be evaluated in fast-casual restaurants.

## Figures and Tables

**Table 1 nutrients-13-02329-t001:** Staging algorithm scoring (after Curry et al., 1992).

Stage of Change	Question Answers
1. Pre-contemplation	Participants are not currently limiting their calorie amount in their diets, and did not think about it over the past month.
2. Contemplation	Participants are not currently limiting their calorie amount in their diets, but are thinking about it and have a little confidence in changing their current diet in the next month.
3. Preparation	Participants are not currently limiting their calorie amount in their diets, but are thinking about it and are somewhat confident in their ability to start to control their diet in the next month.
4. Action	Participants are currently limiting their calories in their diets, and have been doing so for less than 6 months.
5. Maintenance	Participants are currently limiting their calories in their diets, and have been doing so for at least 6 months.

**Table 2 nutrients-13-02329-t002:** Sample demographics (*n* = 471).

	Responses	*n*	%
Gender	Male	272	57.7
	Female	194	41.2
	Other	5	1.1
Race/Ethnicity	Asian	62	13.2
	African American or Black	34	7.2
	Hispanic or Latino	35	7.4
	Multiracial	23	4.9
	Native American or American Indian	2	0.4
	White	315	66.9
Education	High school or equivalent	67	14.2
	Some college credit, no degree	147	31.2
	Vocational training/technical school	11	2.3
	Associate degree	68	14.4
	Bachelor’s degree	161	34.2
	Master’s degree	17	3.6
BMI	Underweight	23	4.9
	Normal	266	56.5
	Overweight	114	24.2
	Obese	68	14.4

**Table 3 nutrients-13-02329-t003:** Subjects’ restaurant choices and random menu assignments.

	Menu Type	*n*	%
Urban Mexican	Control Menu	55	11.7
	First and Last ^a^	53	11.3
	Light and Fresh ^b^	54	11.5
	Sweet Spot ^c^	51	10.8
	Total	213	45.2
Asian Fusion	Control Menu	40	8.5
	First and Last	39	8.3
	Light and Fresh	37	7.9
	Sweet Spot	37	7.9
	Total	153	32.5
California Cafe	Control Menu	28	5.9
	First and Last	23	4.9
	Light and Fresh	28	5.9
	Sweet Spot	26	5.5
	Total	105	22.3

Menus designs based on ^a^ serial position effect (low calorie entrees placed in top-right corner), ^b^ gaze motion (low calorie entrees placed as the first and last items), ^c^ and salience builder effect (icon next to low calorie entrees).

**Table 4 nutrients-13-02329-t004:** Counts for subjects by the independent variables: restaurant type, menu type, stage of change, body mass index, sex, education, and race/ethnicity. Crosstabulations and percentages between the independent variables and the dependent variable menu entree calorie choice (<600 calories or ≥600 calories).

Independent Variable		N		Total	%
		<600 Calories	≥600 Calories		<600 Calories
Restaurant Type	Menu Type				
Urban Mexican	Control Menu	15	40	55	27.3
	First and Last	19	34	53	35.8
	Light and Fresh	27	27	54	50
	Sweet Spot	12	39	51	23.5
	Total	73	140	213	34.3
Asian Fusion	Control Menu	10	30	40	25
	First and Last	15	24	39	38.5
	Light and Fresh	10	27	37	27
	Sweet Spot	15	22	37	40.5
	Total	50	103	153	32.7
California Cafe	Control Menu	13	15	28	46.4
	First and Last	6	17	23	26.1
	Light and Fresh	15	13	28	53.5
	Sweet Spot	14	12	26	53.8
	Total	48	57	105	45.7
Stage of Change	Pre-contemplation	68	158	226	30.1
	Contemplation	12	21	33	36.4
	Preparation	32	53	85	37.6
	Action	41	44	85	48.2
	Maintenance	18	24	42	42.9
BMI	Underweight	14	9	23	60.9
	Normal	92	174	266	36.7
	Overweight	38	76	114	33.3
	Obese	27	41	68	39.7
Sex	Female	89	105	194	45.9
	Male	82	190	272	30.1
	Others	0	5	5	0
Education	High school	23	44	67	34.3
	Some college credit	45	102	147	30.6
	Vocational training/technical school	5	6	11	45.5
	Associate degree	26	42	68	38.2
	Bachelor’s degree	63	98	161	39.1
	Master’s degree	9	8	17	52.9
Race/Ethnicity	Asian	20	42	62	32.3
	Black/African American	11	23	34	32.4
	Hispanic/Latino	13	22	35	37.1
	Multiracial	11	12	23	47.8
	Native American or American Indian	0	2	2	0
	White	116	199	315	36.8
I Am Hungry Right Now	Strongly Disagree	15	30	45	33.3
	Disagree	50	96	146	34.2
	Neither Agree nor Disgree	63	108	171	36.8
	Agree	20	32	52	38.5
	Strongly Agree	23	34	57	40.3
Total Subjects		171	300	471	

**Table 5 nutrients-13-02329-t005:** Final logistic regression (*n* = 471) results from modeling entree calorie choices (< or ≥600 calories) using the predictors found to be significant (*p* < 0.05): menu design, stage of change, sex, and body mass index (BMI).

	B	Df	Sig.	Exp(B)
Menu Design ^a^		3	0.138	
First and Last	0.038	1	0.896	1.039
Light and Fresh	0.594	1	0.036	1.811
Sweet Spot	0.244	1	0.396	1.276
Stage of Change ^b^		4	0.034	
From Stage 1 to 2	0.440	1	0.287	1.552
From Stage 2 to 3	0.118	1	0.703	1.126
From Stage 3 to 4	0.649	1	0.019	1.913
From Stage 4 to 5	−0.038	1	0.913	0.962
Sex ^a^		2	0.015	
Female to Male	0.604	1	0.004	1.830
Other to Male	−20.208	1	0.999	0.000
Body Mass Index ^b^		3	0.132	
Underweight to Normal	−1.032	1	0.024	0.356
Normal to Overweight	−0.623	1	0.054	0.537
Overweight to Obese	−0.218	1	0.486	0.804
Constant	−7.027	1	0.999	0.001

^a^ The simple contrast function was used to compare each level of the menu-type to the control menu (i.e., reference). Sex was evaluated using the simple contrast. ^b^ The difference contrast function was used to compare each level of the stages of change and body mass index to the previous category in the model (except the first one).

## Data Availability

The data presented in this study are available on request from the corresponding author and is available in the publicly accessible repository at https://doi.org/10.6084/m9.figshare.14919864.v1.

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
