# Peer review of "Menu Engineering and Dietary Behavior Impact on Young Adults’ Kilocalorie Choice"

_nutrients, 2021, doi:10.3390/nu13072329_

Round 1

Reviewer 1 Report

FORMAT

  • Be carefull with format, specially with margins

INTRODUCTION

  • Explain better this afirmation: Seaberg first introduced the concept of menu design psychology, and various menu design proposals, also known as menu engineering (lines 85-86). What is Seaberg exactly? Looks like a format not an author
  •  Has this study any hyphotesis?

MATERIALS AND METHODS

  • More specific explanations. Congratulations

RESULTS

  • To understand better the result I recommend to divide sections according to objectives

DISCUSSION

  • Good description

CONCLUSIONS

- There are short: wit this afirmation " Green symbols along with numeric values on menus were associated with young adults that made lower Calorie entrée choices. Subjects who  just changed to a healthier diet over the past six months or less had greater odds of choosing lower Calorie food items; the same was true  for those with a lower BMI" It is necessary to explain more questions: How is understood the menu?, How the menu has to be designed so that nutrition information is understood?, Shoud goverments should legislate in this regard?...

Author Response

Please find below the author's comments related to reviewer #1 recommendations. Our responses are included in a purple-colored font.

FORMAT

  • Be carefull with format, specially with margins.
    • We didn’t change anything in the formatting of the margins unless it happened by accident. Thus, we don’t know what to change to improve it.

INTRODUCTION

  • Explain better this afirmation: Seaberg first introduced the concept of menu design psychology, and various menu design proposals, also known as menu engineering (lines 85-86). What is Seaberg exactly? Looks like a format not an author
    • We changed the sentence to the following to make it clearer.
    • The author Seaberg22 first introduced the concept of menu design psychology, and various menu design proposals, also known as menu engineering. 
  • Has this study any hyphotesis?
    • We added the following to the text.
    • The study hypothesis is as follows: young adults will be positively influenced by several menu engineering formats to choose lower Calorie entrées from a simulated fast-casual restaurant menu. In addition, several demographic and physical characteristics of young adults will also be associated with them choosing lower Calorie entrée choices.

MATERIALS AND METHODS

  • More specific explanations. Congratulations

RESULTS

  • To understand better the result I recommend to divide sections according to objectives
  • We added the following to divide the results into sections.
    • Demographics
    • Menu-Type
    • Menu Design and Calorie Choice
    • Stages of Change and Calorie Choice
    • Demographics and Calorie Choice
    • Degree of Hunger and Calorie Choice

DISCUSSION

  • Good description

CONCLUSIONS

- There are short: wit this afirmation " Green symbols along with numeric values on menus were associated with young adults that made lower Calorie entrée choices. Subjects who  just changed to a healthier diet over the past six months or less had greater odds of choosing lower Calorie food items; the same was true  for those with a lower BMI" It is necessary to explain more questions: How is understood the menu?, How the menu has to be designed so that nutrition information is understood?, Shoud goverments should legislate in this regard?.

  • We aren’t sure what the reviewer is suggesting that we change about the conclusions section. What it seems to be is that they want us to add some of our discussion into the conclusion. We choose to summarize the study’s findings and finish with a sentence about future research needs. If there is something else specific that needs to be changed we will be happy to do so.

Christine Bergman

Reviewer 2 Report

This is a quite interesting intervention study based on menu engineering in out-of-home meals studying how labelling meals may help in choosing healthy options, according to different personal variables.

Some minor comments:

  1. I’d suggest to use an structured format in the abstract (introduction, M&M; results, conclusion)
  2. Introduction: lines 56-59. This sentence is not based on real world data but a wishing thought.
  3. Design
    1. Although it is quoted as a limitation in the discussion section, there is a need to emphasize that it was on on-line experiment, where no real-word restaurant and menus were tested.
  4. Results
    1. Table 1 ref (line 301, 308) should be table 2.
    2. Results section should not have references.
    3. Lines 316-320. This sentence should be moved to discussion section (Limitation of the study).
    4. Table 2. Gender Other was not defined. I suggest no report as NK o Unreported.

Lines 400-403 should be moved to discus

Author Response

Please find below our responses to the comments offered by Editor #2 

Some minor comments:

  1. I’d suggest to use an structured format in the abstract (introduction, M&M; results, conclusion)

We did use a structured format but we didn’t use headings, as is mentioned to do in Nutrients Instructions for Authors.                                                 

  1. Introduction: lines 56-59. This sentence is not based on real world data but a wishing thought.

We modified this sentence to take out the association between increased people eating at restaurants and obesity rates. The sentence now reads as follows: Therefore, food consumed that is prepared at restaurants, will likely continue to increase globally14.

  1. Design
    1. Although it is quoted as a limitation in the discussion section, there is a need to emphasize that it was on on-line experiment, where no real-word restaurant and menus were tested.

We also used the word “simulated” and “online environment” to address this issue. Please see the following:

The study hypothesis is as follows: young adults will be positively influenced by several menu engineering formats to choose lower Calorie entrees from a simulated fast-casual restaurant menu.

All the Calorie information, menu items and descriptions were obtained from three fast-casual restaurant chains, considered the models for this simulation study.

A limitation of this study is that the menu selection was conducted in an online environment instead of in a fast-casual restaurant. The use of the online environment can introduce sampling bias in that all members of the target population do not have equal access to this format.

These results from this simulation study need to be evaluated in fast-casual restaurants.

  1. Results
    1. Table 1 ref (line 301, 308) should be table 2.

This mistake was corrected.

    1. Results section should not have references.

We removed this paragraph with the reference related to sample size and added it to the methods section, in the first paragraph.

    1. Lines 316-320. This sentence should be moved to discussion section (Limitation of the study).

The discussion section already included a limitation related to education. Please see the following sentence: The population in this study and in other studies using online survey platforms were more educated than the U.S. population overall35.

We added an additional sentence to address the limitation related to the race of the sample members as follows:

We added the following sentence to address the reviewer's concern. In addition, the sample for this study was different from the U.S. population in terms of race.

We also removed the mention of a limitation from the results section as suggested by the reviewer.

    1. Table 2. Gender Other was not defined. I suggest no report as NK o Unreported.

We would like to leave this data in the table as Gender Other had 5 subjects. We think other is self-explanatory – any gender other than male or female. For example, some people are born with both sex attributes of both men and women. Also, we didn't define male or female so we don't think it is necessary to define other. Also, 5 subjects responded as other. However, if this is a problem we will definitely remove the data.

Lines 400-403 should be moved to discus

We added this sentence to the end of the following paragraph which is the first paragraph in the discussion section.

According to the U.S. Census Bureau, in 2012, those aged from 18-24 were segmented into the top four groups by race/ethnicity as follows: Caucasian (56%), Asian (5%), Hispanic or Latino (19%) or African American or black (15 %). Therefore, the sample used for this study over-represents Caucasians and Asians and underrepresents Hispanics (or Latinos) and African Americans (or blacks).38  In addition, the sample in this study was also different in terms of education level compared to the U.S. population of young adults.

Many thanks for your help improving our paper,

Christine Bergman